# Initial Response of Pentaerythritol Tetranitrate (PETN) under the Coupling Effect of Preheating, Shock and Defect via the Molecular Dynamics Simulations with the Multiscale Shock Technique Method

**DOI:** 10.3390/molecules28072911

**Published:** 2023-03-24

**Authors:** Yaping Zhang, Tao Wang, Yuanhang He

**Affiliations:** 1Institute of Intelligent Manufacturing Technology, Shenzhen Polytechnic, Shenzhen 518055, China; 2Shenzhen Institutes of Advanced Technology, Chinese Academy of Sciences, Shenzhen 518055, China; 3State Key Laboratory of Explosion Science and Technology, Beijing Institute of Technology, Beijing 100081, China

**Keywords:** PETN, coupling effect, preheating, MSST

## Abstract

The initial response of PETN under the coupling of preheating, impact and defects was simulated by Multiscale Shock Technique (MSST) method and molecular dynamics. The temperature change of PETN during impact compression can be divided into three stages: (1) the elastoplastic change of the system caused by initial compression; (2) part of PETN decomposes and releases energy to raise temperature; (3) a secondary chemical reaction occurs, resulting in rapid temperature rise. Under the given conditions, a higher initial preheating temperature will lead to faster decomposition of PETN; The existence of defects will accelerate the decomposition of PETN molecules; Coupling the highest preheating temperature with defects will lead to the fastest decomposition of PETN molecules, while in the defect-free PETN system with a preheating temperature of 300 K, the decomposition of PETN molecules is the slowest. For the case of *U_s_* = 8 km·s^−1^, the effect of defects on the initial PETN reaction is greater than the initial preheating temperature; When the impact velocity is greater than 9 km·s^−1^, the impact velocity is an important factor affecting the decomposition of PETN molecules. For *U_s_* = 10 km·s^−1^, NO_2_ is the main initial product in the defective PETN crystal, while in the perfect PETN crystal, it is the combination of NO_2_ and HONO. The chemical reaction kinetics analysis shows that the preheating temperature and defects will accelerate the decomposition of PETN. The higher the preheating temperature, the faster the decomposition of PETN. For the case of *U_s_* = 7 km·s^−1^, 8 km·s^−1^ and 9 km·s^−1^, the existence of defects will increase the decomposition rate by more than 50% regardless of the initial preheating temperature. In the case of *Us* = 10 km·s^−1^, the improvement of decomposition rate by defects is not as significant as the initial preheating temperature.

## 1. Introduction

The study of shock initiation of energetic materials is one of the core contents of applied detonation physics, but it is difficult to explain the shock initiation mechanism of energetic materials from the atomic and molecular level by current experiments and theoretical methods. In practical applications, the response of energetic materials to external stimuli is very complex because a variety of external stimuli act on energetic materials at the same time. Recently, scholars have begun to pay attention to the coupling effect of multiple external stimuli on the response mechanism [1,2], especially the influence of thermal shock coupling on the safety of energetic materials [3,4].

When the energetic material RDX was preheated, Urtiew [3]. Dreger et al. [5,6] thought that preheating improved the chemical reaction rate of RDX, which increased the shock sensitivity of RDX. Akiki et al. [7] proposed a hot spot formation model based on inhomogeneous effects of physical and chemical properties and used it to predict the reaction of energetic materials under different preheating temperatures. In addition, crystal defects (such as twins, molecular vacancies, and dislocations) [8,9,10] in energetic materials would also have a significant shock on the safety of energetic materials. Kuklja et al. [11] proved by the ab initio method that vacancies and voids accelerate the thermal decomposition of HMX by increasing surface area. Zhou et al. [12] studied the high-temperature thermal decomposition mechanism of condensed phase HMX containing molecular vacancies by the method of molecular dynamics simulation of the ReaxFF reaction force field and pointed out that molecular vacancies significantly promoted intramolecular reactions, such as the fracture of N-NO_2_ bond, the dissociation of HONO and the fracture of C-N main ring. Wen et al. [13] considered that under the same shock conditions, the twins significantly improved the shock sensitivity of HMX, and the hot spots were easy to form at the twins.

The most common state of energetic materials is that they are subjected to multiple stimuli at the same time, and there are more or fewer defects in energetic materials. Coupling these external stimuli with crystal defects will increase the difficulty of revealing the safety mechanism. In addition to experimental measurements and theoretical derivation, molecular dynamics simulation has been proven to be an effective tool for revealing the atomic and molecular details of energetic material systems under external stimuli. Wood et al. [14] pointed out that the decomposition of energetic materials can be induced at relatively low energy if appropriate thermal and electromagnetic coupling methods are adopted. Xue et al. [15] used the ReaxFF reaction force field Molecular Dynamics (MD) method to study the influence of dislocations on the shock sensitivity of RDX and found that edge dislocations significantly improved the shock sensitivity. Wang et al. [16] explained the thermal decomposition of RDX on the surface of nano-aluminum particles. The results showed that RDX could spontaneously decompose on the surface of aluminum due to the strong interaction between aluminum and oxygen atoms. Zhong et al. [17] used the ReaxFF reaction force field molecular dynamics method to simulate the influence of common components in the atmosphere on the thermal decomposition of RDX and found that CO_2_, H_2_O, H_2_, and N_2_ inhibited the decomposition of RDX to some extent. Sharia et al. [18] studied the effect of defect induction on the surface decomposition of HMX molecular crystals and pointed out that changing the crystal structure can effectively control the decomposition of HMX molecules. Kimmel et al. [19] introduced the influence of charge and excited state on the decomposition of 1, 1-diamino-2, and 2-dinitroethylene (FOX-7). The results showed that when there were free electrons near FOX-7 molecules, the formation of HONO was a feasible exothermic reaction with a relatively low energy barrier. Kuklja et al. [20,21,22] used the DFT method to study the influence of shear strain on the activation barrier of nitroaromatic hydrocarbons. The results showed that the interaction between initial chemical products and molecular crystals would increase shear strain and accelerate the decomposition process.

At present, there are few studies on energetic materials under the coupling of various loads. Understanding the initial response of energetic materials under the coupled action of preheating, shock and defects can provide important theoretical significance for the transportation and storage of energetic materials. The initial response of perfect PETN and PETN with defects under the combined action of heat and shock would be simulated, and the effects of preheating, shock and defect on the initial response of energetic materials would be further discussed.

## 2. Results and Discussion

### 2.1. Thermodynamic Quantities during Shock Compression

MSST, combined with the molecular dynamics simulation method, can accurately predict the thermodynamic quantities of the chemical reaction zone in front of the wavefront or behind the wavefront of energetic materials during shock compression. Figure 1 is the evolution curve of temperature with time during shock compression. Looking at Figure 1, it can be seen that the temperature change can be divided into three stages: (1) during initial compression (within 1 ps), the temperature rises sharply, which corresponds to the elastic-plastic change of the PETN crystal under the action of a shock wave; (2) After 1 ps, some PETN decomposed with the increase of system temperature, and the energy released further increased the temperature; (3) The occurrence of secondary chemical reaction makes the temperature rise slowly. Comparing Figure 1a–c, it can be seen that the temperature change of the defect-free PETN system is not obvious at the shock speed of 7 km·s^−1^; that is, the initial compression only causes the elastic change of the PETN system; At the case of *U_s_* = 8 km·s^−1^, the initial preheating temperature can significantly change the temperature of PETN system, and the temperature of PETN system with the initial preheating temperature of 700 K reaches 4500 K during shock compression, which exceeds the CJ point temperature of PETN; At the cases of *U_s_* = 9 km·s^−1^ and 10 km·s^−1^, the initial thermal bath temperature has little effect on the temperature of PETN system.

Comparing the defect-free PETN system with the defect-containing PETN system, the temperature of the defect-containing PETN system is always higher than that of the defect-free PETN system under the same initial preheating and shock conditions. Observing Figure 1a–c, it is found that the initial preheating temperature has an obvious influence on the temperature change of the PETN crystal during shock compression in the case of *U_s_* = 7 km·s^−1^. At the end of the simulation time, the temperature of the PETN system with the initial preheating temperature of 300 K is 1800 K, while the temperature of the PETN system with the initial preheating temperature of 700 K is as high as 4200 K; At the cases of *U_s_* = 8 km·s^−1^, 9 km·s^−1^ and 10 km·s^−1^, the initial thermal bath has little effect on the temperature change of PETN system.

The pressure produced in the process of shock compression can also be used as one of the parameters to measure the reaction degree and sensitivity. Figure 2 shows the evolution curve of pressure with time during shock compression. The pressure remains stable shortly after the shock loading, and the pressure depends on the shock speed to a great extent; that is, the greater the shock speed, the higher the pressure.

Comparing Figure 2a–c, it can be seen that the initial preheating temperature has little influence on the pressure. Similarly, defects have little influence on the pressure of the PETN crystal. Therefore, it is difficult to use pressure as an index to explain the influence of initial preheating on shock sensitivity. The change of pressure can be divided into three stages: (1) pressure rise caused by initial shock compression within 1 ps; (2) the pressure rise caused by the chemical reaction within 1~5 ps; (3) after 5 ps, the pressure decreases slowly, which is due to the expansion of the medium in the reaction process.

### 2.2. Relationship between Shock Wave Speed and Particle Speed

Under the action of the shock wave, there is a linear relationship between the shock wave speed of energetic materials and the particle speed on the wavefront, that is,
(1)$$U_{s}=C_{0}+SU_{p}$$
where $C_{0}$ is the sound speed of materials under zero pressure, and S is the adiabatic bulk modulus. Figure 3 shows the relationship between PETN shock wave speed and particle speed. It can be seen from Figure 3 that the particle speed *U_p_* obtained by linear fitting is between 2.14 km·s^−1^ to 4.62 km·s^−1^.

In Figure 3, the straight line obtained by fitting Equation (1) shows that the initial preheating temperature has little effect on the relationship between shock wave speed and post-wave particle speed. For the PETN system with an initial thermal bath temperature of 300 K, the speed relations for the system with and without defects are as follows: *U_s_* = 3.8 + 1.49 *U_p_* and *U_s_* = 1.92 + 1.83 *U_p_*. Obviously, the sound velocities of the fitted materials are obviously different, which may be due to the different densities of the two PETN systems after optimization. The initial density of perfect PETN after optimization is 1.789 g·cm^−3^, while that of PETN with defects after optimization is 1.626 g·cm^−3^. The speed relation of perfect PETN is close to the *U_s_* = 3.7 + 1.56 *U_p_* calculated by Shan et al. [23].

Combined with the Rankine–Hugoniot relation, the shock initiation pressure of PETN can be calculated by Equation (2).
(2)$$P=\rho_{0}U_{s}U_{p}$$
where $\rho_{0}$ is the initial density of the material, 1.789 g·cm^−3^ is taken in this paper; *U_s_* is the minimum shock wave speed to excite the chemical reaction of PETN, and *U_p_* is the particle speed corresponding to *U_s_*. The calculated pressure for *Us* = 7 km·s^−1^, 8 km·s^−1^, 9 km·s^−1^ and 10 km·s^−1^ is 27.29 GPa, 40.07 GPa, 54.74 GPa and 75.13 GPa, respectively, which is close to the experimental results of 35.81 GPa, 42.70 GPa, 58.54 GPa and 78.31 GPa of Dick et al. [24,25].

### 2.3. Effects of Initial Preheating Temperature and Defects on PETN Reactivity

Figure 4 is the time evolution curve of the PETN molecular decomposition process during shock compression. Obviously, higher shock wave speed leads to faster decomposition of PETN molecules: In the case of *U_s_* = 7 km·s^−1^, PETN molecules in defect-free systems are not completely decomposed in simulation time, while PETN molecules in defect-containing systems are completely decomposed in simulation time; At the case of *U_s_* = 8 km·s^−1^, the decomposition of PETN in all systems is complete within 5~20 ps, except that in the defect-free PETN system at 300 K; At the case of *U_s_* = 9 km·s^−1^, PETN molecule decomposes completely at 12 ps; At the case of *U_s_* = 10 km·s^−1^, PETN molecule decomposes completely at 4 ps. It should be reasonable because a higher shock speed has a higher ability to decompose PETN.

Looking at Figure 4b and Figure 5a, it can be seen that defects have a greater influence on PETN molecular decomposition than the initial preheating temperature at *U_s_* = 7 km·s^−1^ and 8 km·s^−1^. In the case of *U_s_* = 8 km·s^−1^, the PETN molecules in the defect-free PETN system with an initial thermal bath temperature of 700 K decompose completely at 13 ps, while the PETN molecules in the defect-containing PETN system with an initial thermal bath temperature of 300 K decompose completely at 9 ps.

When the shock velocities are *U_s_* = 9 km·s^−1^ and 10 km·s^−1^, the effects of the initial thermal bath and defects on the reactivity of PETN molecules are not significantly different, especially at *U_s_* = 10 km·s^−1^, the PETN molecules of each system decompose rapidly and completely, and the time difference is less than 2 ps.

Under given conditions, the higher initial preheating temperature would lead to faster decomposition of PETN; The existence of defects would accelerate the decomposition of PETN molecules. In addition, coupling the highest preheating temperature with defects will lead to the fastest decomposition of PETN molecules, while at 300 K preheating temperature, the decomposition of PETN molecules in defect-free PETN crystals is the slowest.

### 2.4. Initial Decomposition Path and Product Distribution of PETN

PETN would decompose continuously during shock compression to form new molecules and molecular groups. From the above analysis, it can be seen that in the case of *U_s_* = 7 km·s^−1^, most PETN molecules in the defect-free PETN system are not completely decomposed; While the shock speed is greater than 9 km·s^−1^, the shock speed has the greatest influence on the decomposition of PETN molecules. Therefore, the shock condition of *U_s_* = 8 km·s^−1^ is selected as the research object in this section. Figure 5 shows the evolution curve of the initial decomposition products of PETN with time.

At the same shock speed, defects significantly affect the formation of initial decomposition products. In PETN, the fracture of the O-NO_2_ bond to NO_2_ is considered as the initial reaction channel. In Figure 5, NO_2_ and C_5_O_10_N_3_H_8_ always appear first in the reaction products, and their quantities gradually increase with the increase of initial preheating temperature. Compared with Figure 5a,b, the initial decomposition products of the PETN crystal with defects are more than those of the perfect PETN crystal, and the chemical reaction reaches equilibrium earlier at the preheating temperature of 300K. In Figure 5a, the number of various products still increased because some PETN in the system did not decompose at 20 ps. In Figure 5b, the amount of NO_2_ reaches the maximum value of 255 at 9 ps, and then NO_2_ continuously participates in the chemical reaction as a reactant to form other stable small molecular products. NO, HONO, and HNO_3_, which formed after NO_2_, also reached the maximum at about 10 ps and then participated in the subsequent redox reaction to form stable products, such as N_2_, H_2_O, and CO_2_.

Comparing Figure 6a,c,e, it can be found that the increase of initial preheating temperature stimulates the chemical reaction in a defect-free PETN system, and in Figure 5e, the amount of NO_2_ reaches the maximum at 10 ps, and the amount of other intermediate products reaches the maximum between 10 and 15 ps. Comparing Figure 5b,d,f, it is found that the initial preheating temperature does not obviously promote the chemical reaction in the PETN crystal with defects, and the values of NO_2_, NO, HONO, HNO_3_ and other products reach the maximum between 5~10 ps, and then decrease slowly. It is also verified that the shock speed *U_s_* = 8 km·s^−1^ has a greater influence on the initial response of PETN than the initial thermal bath temperature.

Figure 6 shows the evolution curve of main reaction products with time at *U_s_* = 8 km·s^−1^. In Figure 6a, NO_2_ is formed as the initial product of the reaction with the decomposition of PETN. By comparing with Figure 4b, the peak time of NO_2_ is almost consistent with the complete decomposition time of PETN. Except for the perfect systems with initial preheating temperatures of 300 K and 500 K, the NO_2_ in other systems would experience a short equilibrium period after reaching the peak value; that is, the production and consumption of NO_2_ in the system are in dynamic equilibrium in a short time.

In Figure 6c,e,f, NO, HONO and HNO_3_, as intermediates of the reaction, also show similar changes to NO_2_. However, the change of NO_3_ has no obvious law, and its amount is far less than that of NO_2_ and other products.

N_2_, CO_2_, H_2_O, etc., as the final products of the reaction, only appear under specific conditions. In the defect-free system, only a small amount of H_2_O was formed, and no N_2_ or CO_2_ was formed at the initial thermal bath temperature of 300 K. Comparing Figure 6b,g,h, it can be seen that H_2_O formed earlier than N_2_ or CO_2_, and the amount of H_2_O was more than that of N_2_ or CO_2_. Under the shock condition of *U_s_* = 8 km·s^−1^, the formation of CO was not observed, which may be due to the short simulation time. Li et al. [26] thought that when the reaction zone was extended to the ns level, a large number of carbon oxides, especially CO, would be formed.

### 2.5. Initial Response of PETN under Strong Load Coupling

Dlott et al. [27,28] believed that even in the absence of defects, it was possible to initiate spontaneous ignition of energetic materials as long as there was enough high energy and long action time. Therefore, this section focuses on the initial response of PETN under strong load (*U_s_* = 10 km·s^−1^) coupling. Figure 7 shows the evolution curve of initial decomposition products with time at *U_s_* = 10 km·s^−1^.

In the previous analysis of the initial decomposition products at *U_s_* = 8 km·s^−1^, it is considered that the defects significantly affect the formation of the initial decomposition products at the same shock speed. Looking at Figure 7, it can be seen that due to the high shock speed, defects have little effect on the number of initial products except for NO_2_; The quantity of all initial products does not increase much with the increase of initial preheating temperature, and the quantity of nitrogen dioxide is always more than that of other nitrogen oxides. Compared to Figure 7a,b, the peak NO_2_ amount in the system with defects was 1.5 times higher, and the peak NO_2_ amount was reached earlier at a preheating temperature of 300 K than in the system without defects. This is due to the existence of defects, which increase the reaction channels of PETN decomposition. It is possible that individual PETN loses two or even three NO_2_ at the same time during the decomposition process. Similar results were found when studying the binary collision process of PETN by Wu et al. [29].

Comparing Figure 7a,c,e, the amount of each initial nitrogen oxide does not change significantly. When the amount of NO_2_ reaches its peak value, it will go through an equilibrium period of about 2~3 ps, and then rapidly decrease to zero. However, in the system with defects, there is no such equilibrium period in the evolution of NO_2_ quantity, and the amount of NO_2_ decreases to zero within 1 ps after reaching the peak. The maximum concentration ratios of NO_2_ to HONO, NO, HNO_3_ and NO_3_ were 2.31, 4.11, 3.7, and 7.4 in perfect PETN. The maximum concentration ratios of NO_2_ to HONO, NO, HNO_3_ and NO_3_ were 3.61, 3.81, 9.5, and 9.1 in the PETN system with defects. The results show that NO_2_ is the main initial product in the defect-containing system, while NO_2_ and HONO act together in the defect-free system. Although the initial decomposition path of PETN is NO_2_ formed by the fracture of O-NO_2_ bond because the reaction of eliminating HONO is exothermic, and the heat released by it is used to support the reaction and accelerate the chemical reaction process [30], it is reasonable for NO_2_ and HONO to be the initial reaction products together.

### 2.6. Chemical Reaction Kinetics

Energy is an important factor that determines PETN attenuation during shock compression. In particular, the reactivity is determined by the potential energy at a given temperature; that is, the higher the potential energy, the higher the reactivity of the same chemical composition of PETN. The evolution of potential energy under various conditions is shown in Figure 8. As can be seen from Figure 8, the potential energy of the system decreases rapidly in the initial stage of the reaction. Because the total energy of the system remains unchanged, the reduced potential energy increases the kinetic energy of the system, which is in the induction stage at this time. With the progress of the reaction, secondary reactions, mainly exothermic reactions, began to occur, and complex intermediate products began to form and finally transform into stable small molecular structures. Due to a large amount of heat released in the reaction process, the energy of the system gradually decreases until the reaction equilibrium.

In the case of *U_s_* = 7 km·s^−1^, the potential energy of all PETN systems has no obvious change after the induction period, which is due to the fact that there are few chemical reactions and no secondary chemical reactions in the PETN crystal at this shock speed. Looking at Figure 8a–c, when the shock velocities *U_s_* = 9 km·s^−1^ and 10 km·s^−1^, due to the complexity of chemical reactions in the system, the change of potential energy of the system goes through three stages: (1) After the induction period ends, the potential energy enters the equilibrium period; (2) The secondary chemical reaction causes the potential energy to drop rapidly to the lowest value; (3) Because of the large amount of energy released in the reaction process, the potential energy of the system increases slowly.

Compared with Figure 8a–c, defects shorten the equilibrium period of potential energy. After a short induction period, the potential energy of the system continues to decrease, but the rate is lower than that during the induction period. After the potential energy drops to the lowest value, it gradually rises.

In Figure 8, it is found that the main position of the potential energy curve is distinguished by the initial preheating temperature; that is, a higher preheating temperature leads to higher shock resistance potential energy. Therefore, this higher potential energy makes the decomposition reaction faster in any case, such as the rapid reduction of potential energy or potential energy release shown in Figure 8. Due to the higher initial potential energy and the faster potential energy reduction, the intersection of potential energy curves appears in Figure 8c.

After the initial equilibrium and induction period, the potential energy of the system decays with the exothermic chemical reaction, and PETN molecules begin to decompose. In order to quantitatively describe the decomposition rate of PETN, the first-order exponential function [12] is used to fit the decomposition process of PETN molecules, and the equation is as Equation (3):(3)$$\alpha \left(t\right)=1-e^{-kt}$$
where α(t) is the reaction progress, that is, the number of PETN molecules in the system/the initial number of PETN molecules at time *t*, and *k* is the decay rate (unit is 10^12^ s^−1^). By fitting Figure 3 with Equation (3), the attenuation rate values of each system can be obtained, as shown in Table 1. Where *k*_p_ and *k*_d_ represent the decomposition rate of PETN molecules in defect-free and defect-containing systems, respectively. In Table 1, both the initial preheating and defects accelerate the decomposition of PETN, and the higher the preheating temperature, the faster the decomposition of PETN. In addition, the dependence of defects on the decomposition rate is verified by calculating the relative increment of *k*_d_ to *k*_p_ at different initial preheating temperatures. From the data in Table 1, this dependence does not seem obvious. According to the data in the table, at *U_s_* = 7 km·s^−1^, 8 km·s^−1^ and 9 km·s^−1^, the presence of defects increases the decomposition rate by more than 50% regardless of the initial preheating temperature. While in the case of *U_s_* = 10 km·s^−1^, the effect of defects on the decomposition rate is not as obvious as that of initial preheating.

In conclusion, the decomposition rate *k* value indicates that the initial preheating and the existence of defects promote the chemical decomposition of shock PETN and also improve the shock sensitivity of PETN.

## 3. Simulation Details

### 3.1. Details of the Multiscale Shock Technique (MSST) Method

Molecular dynamics simulation is used to study the initial shock response of energetic materials. The usual method is to give the whole particle speed of energetic materials and shock the fixed reflector wall to generate a shock wave [31,32,33]. The physical and chemical changes, such as cavity collapse and hot spot formation of energetic materials during shock compression, can be observed by the above method, but this method requires a large simulation system (hundreds of thousands to millions of atoms) and high computer performance. Multiscale Shock Technique (MSST) based on the one-dimensional Euler equation of compressible fluid proposed by Reed et al. [34] can solve the problem of large model size. This technique can not only describe the final state of shock but also determine the instantaneous change of the shock process by capturing the dynamic process of the shock process. The multi-scale shock technique combines molecular dynamics simulation with Euler or compressible Navier-Stokes equations, which can significantly reduce the required model size.

MSST can describe the state evolution before and after the shock wave structure when the shock wave sweeps through the atomic system and can track the change of thermodynamic quantities in the chemical reaction zone. In the field of energetic materials, MSST can accurately describe the evolution of thermodynamic quantities in the front wavefront and the rear chemical reaction zone after the shock wave sweeps through energetic materials through the equation of state in front of and behind the wavefront. At present, MSST combined with molecular dynamics simulation has been widely used in the study of shock compression of energetic materials [13,15,35,36], and the results are in good agreement with the experimental results. More details about the MSST method are shown in Appendix A.

### 3.2. Details of MD Simulation

The basic idea of the molecular dynamics method is that after giving the initial motion state of the molecular system, the natural motion of the molecules is sampled in the phase space for statistical calculations, and variables, such as molecular motion position, velocity and acceleration are calculated according to Newtonian and statistical mechanics methods, and the macroscopic physical quantities, such as pressure, temperature and energy of the whole system are solved from them. Thus, the MD method can not only directly simulate the macroscopic properties of complex multi-atomic systems but also provide the evolution of the relationship between molecular structure and macroscopic properties over time. ReaxFF reaction force field is developed based on the first principle quantum mechanics method, with the bond level as the core, which represents the complex interactions between atoms and atoms in the form of different functions. These functions include bond, angle, dihedral angle, Coulomb, van der Waals and correction terms, and all potential energy terms related to the bond valence depend on the bond level, which is determined by the interatomic distance. ReaxFF reaction force fields can describe the complex changes of interatomic bonds during chemical reactions and thus can accurately describe the breaking of old bonds and the formation of new bonds, providing important information for the study of chemical reaction processes. Through nearly two decades of development, ReaxFF has covered most of the elements in the periodic table, especially in the field of energy-containing material calculations containing C, H, O, and the four elements, and can almost reach an accuracy similar to that of quantum mechanics. Compared to quantum mechanical methods, ReaxFF calculations are faster, the systems studied are larger (up to millions of atoms), the simulation times are longer (over 100 ns), some experimental phenomena can be explained, and it offers the possibility to study the relationship between microscopic properties and macroscopic models of energy-containing materials. In the ReaxFF reaction force field, the charges are calculated using the EEM (Electron Equilibration Method) algorithm, i.e., the instantaneous atomic charges are calculated from the interatomic distance variation for chemical bond breaking and generation in the reaction force field, and all atomic pairs are considered for Coulomb interaction.

Taking PETN as the research object, a 6 × 6 × 9 supercell (55.65 Å × 55.65 Å × 59.51 Å) is constructed, which contains 18,792 atoms. The supercell structure is optimized by energy minimization, and then the system is placed in an NVT ensemble (The canonical ensemble, abbreviated as NVT, represents the number of particles (N), the volume (V) and the temperature (T) with a definite number of particles.) (300 K) for 10 ps, followed by NPT ensemble (The isothermal, isobaric system (constant-pressure, constant-temperature), abbreviated as NPT, represents a system with a defined number of particles (N), pressure (P) and temperature (T).) (300 K, 0 GPa) for 60 ps to optimize atomic coordinates. After optimization, the density of the PETN supercell is 1.789 g·cm^−3^, which is close to the experimental density of 1.773 g·cm^−3^ [37]. At the center of the optimized PETN system, a spherical region with a radius of *R* = 1.6 nm is removed, and PETN crystals with defects are formed. The defect volume accounts for about 9% of the total PETN volume (including about 30 PETN molecules), which is close to the limit of 30% of the observed defect concentration [38].

In order to study the effect of preheating on PETN, the optimized perfect PETN and defect-containing PETN crystals are placed in 300 K, 500 K, and 700 K preheating, respectively. Through the shock loading experiment on the PETN single crystal, Dick et al. [24,25] found that PETN has the lowest shock sensitivity in [100] direction. In order to better describe the initial response of PETN under shock, shock compression along the crystal [100] direction would be carried out. The applied shock wave velocities are 7 km·s^−1^, 8 km·s^−1^, 9 km·s^−1^ and 10 km·s^−1^. The model of PETN supercell and shock compression is shown in Figure 9. Periodic boundary conditions are used in the x, y, and z directions in the simulation process. The compression duration is 20 ps, and the time step is 0.1 fs.

All the above processes are simulated by LAMMPS [39,40] software integrated with MSST and ReaxFF-lg [41] package. The ReaxFF-lg force field has been applied to the shock initiation of energetic materials. More details about the ReaxFF-lg are shown in Appendix A.

## 4. Conclusions

The initial response of PETN under the coupled action of heat shock and the defect was simulated by MSST combined with molecular dynamics. The effects of heat shock and defects on the initial decomposition of PETN were studied. The decomposition path and product distribution of PETN under strong and weak loads were determined. The decomposition rate of PETN was quantitatively described by chemical reaction kinetics.

The temperature change of PETN during shock compression can be divided into three stages: (1) Elastoplastic change of the system caused by initial compression; (2) Part of PETN decomposes and releases energy to raise the temperature; (3) The occurrence of secondary chemical reaction makes the temperature rise rapidly. In the case of *U_s_* = 8 km·s^−1^, the initial preheating can obviously raise the temperature of the PETN crystal; in the case of *U_s_* = 9 km·s^−1^ and 10 km·s^−1^, the effect of initial preheating on the whole system temperature is not obvious. Under all shock velocities, the influence of defects on the temperature of the whole system is limited. Under given conditions, the higher initial preheating temperature would lead to faster decomposition of PETN; The existence of defects would accelerate the decomposition of PETN molecules. Moreover, coupling the highest preheating temperature with defects would lead to the fastest decomposition of PETN molecules during the slowest decomposition of PETN molecules in a defect-free PETN system at 300 K preheating temperature.

When the shock speed *U_s_* = 8 km·s^−1^, the shock of defects on the initial response of PETN is greater than that of the initial thermal bath temperature; However, when the shock speed is greater than 9 km·s^−1^, the shock speed is an important factor affecting the molecular decomposition of PETN. In the case of *U_s_* = 10 km·s^−1^, NO_2_ is the main initial product in the defect-containing system, while NO_2_ and HONO interact together in the defect-free system.

Chemical reaction kinetics analysis shows that both initial preheating and defects accelerate the decomposition of PETN, and the higher the preheating temperature, the faster the decomposition of PETN. At *U_s_* = 7 km·s^−1^, 8 km·s^−1^ and 9 km·s^−1^, the presence of defects increases the decomposition rate by more than 50% regardless of the initial thermal bath temperature. However, in the case of *U_s_* = 10 km·s^−1^, the effect of defects on the decomposition rate is not as obvious as that of initial preheating.

## Figures and Tables

**Figure 1 molecules-28-02911-f001:**
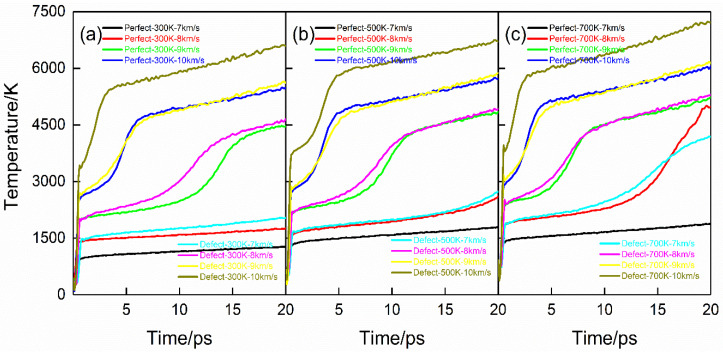
Temperature evolution curves for the shock compression. (**a**–**c**) correspond to PETN crystal with preheating temperatures of 300 K, 500 K, and 700 K.

**Figure 2 molecules-28-02911-f002:**
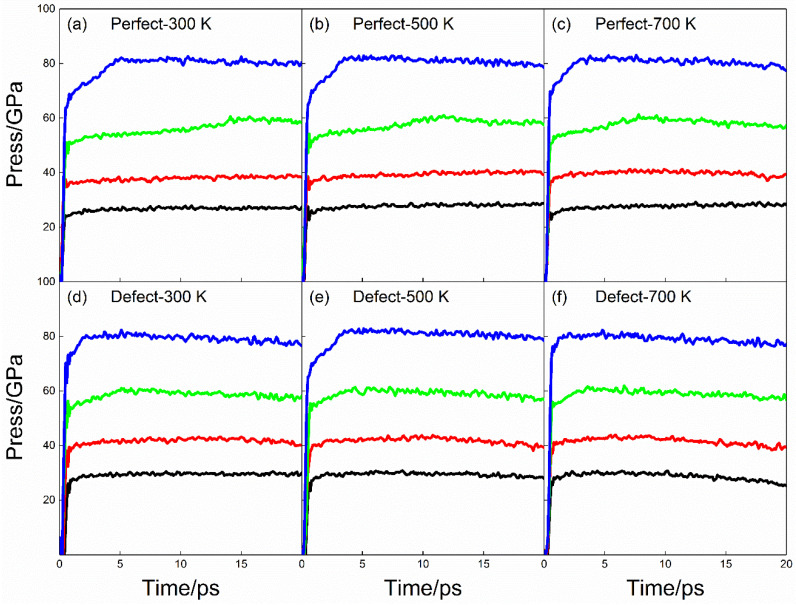
Press evolution curves for the shock compression. (**a**–**c**) correspond to perfect PETN crystal with preheating temperatures 300 K, 500 K, and 700 K; (**d**–**f**) correspond to defective PETN crystal with preheating temperatures 300 K, 500 K, and 700 K; black, red, green, and blue line correspond to shock velocities, *U_s_* = 7 km·s^−1^, 8 km·s^−1^, 9 km·s^−1^, 10 km·s^−1^.

**Figure 3 molecules-28-02911-f003:**
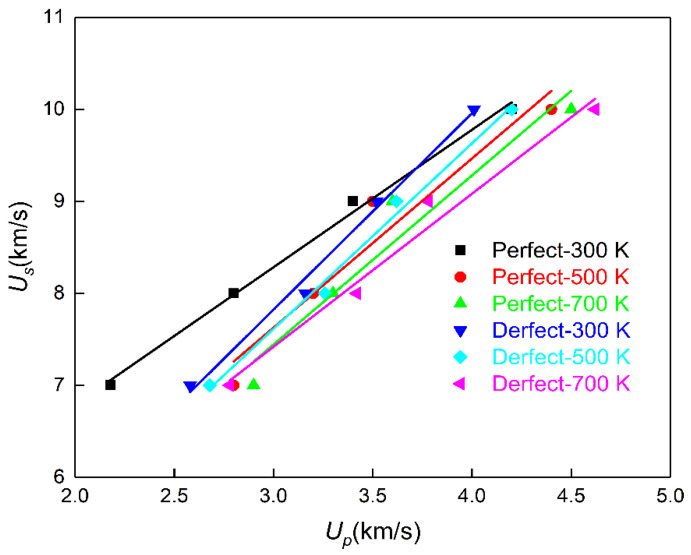
Shock wave velocity versus particle velocity.

**Figure 4 molecules-28-02911-f004:**
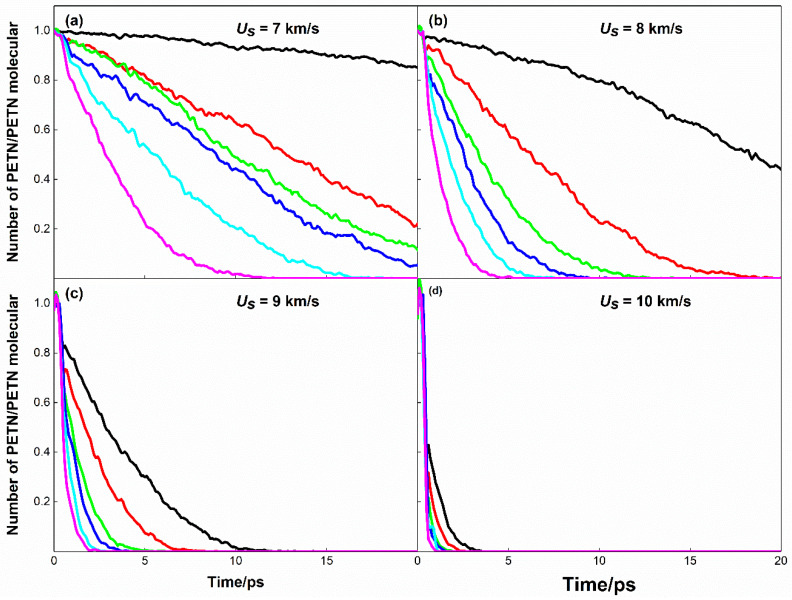
Evolution curves of PETN molecular number during the shock loading. (**a**–**d**) correspond to shock velocities, *Us* = 7 km·s^−1^, 8 km·s^−1^, 9 km·s^−1^, 10 km·s^−1^. The vertical coordinates are normalized by dividing by the number of PETN molecules in the initial system. Black, red, and green lines correspond to perfect crystal with preheating temperatures of 300 K, 500 K, and 700 K. Blue, cyan, and magenta line correspond to Defect crystal with preheating temperatures of 300 K, 500 K, and 700 K.

**Figure 5 molecules-28-02911-f005:**
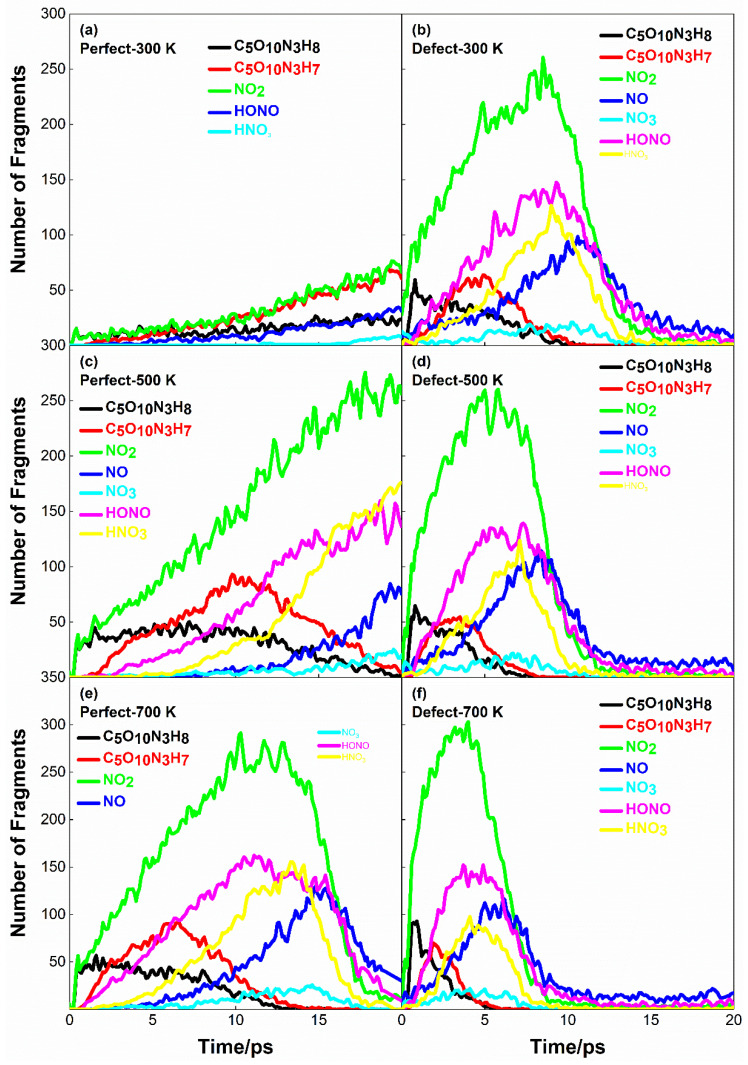
Evolution curves of initial decomposition products for the case of *U_s_* = 8 km·s^−1^. (**a**,**c**,**e**) correspond to perfect PETN crystal with preheating temperatures 300 K, 500 K, and 700 K; (**b**,**d**,**f**) correspond to defective PETN crystal with preheating temperatures 300 K, 500 K, and 700 K.

**Figure 6 molecules-28-02911-f006:**
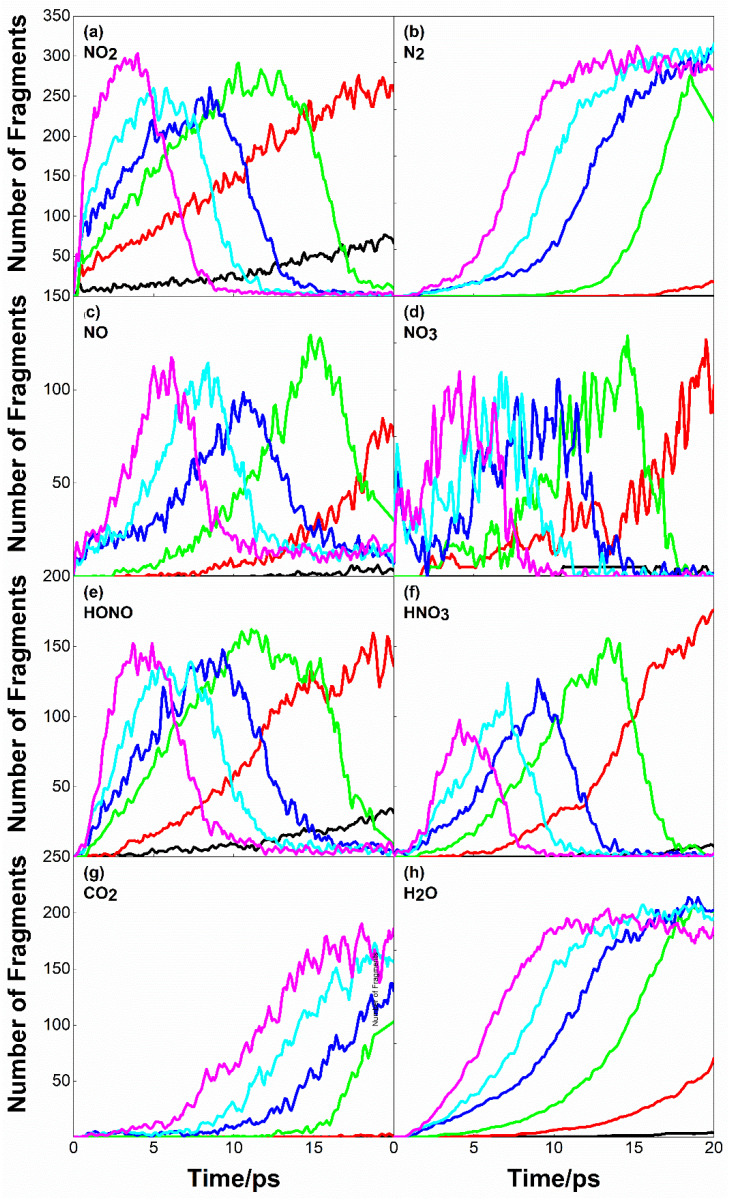
Main reaction products evolution curves for the case of *U_s_* = 8 km·s^−1^. (**a**–**h**) correspond to the products NO_2_, N_2_, NO, NO_3_, HONO, HNO_3_, CO_2_, and H_2_O, respectively. The black, red and green lines correspond to perfect crystal with preheating temperatures of 300 K, 500 K, and 700 K. Blue, cyan, and magenta line correspond to Defect crystal with preheating temperatures of 300 K, 500 K, and 700 K.

**Figure 7 molecules-28-02911-f007:**
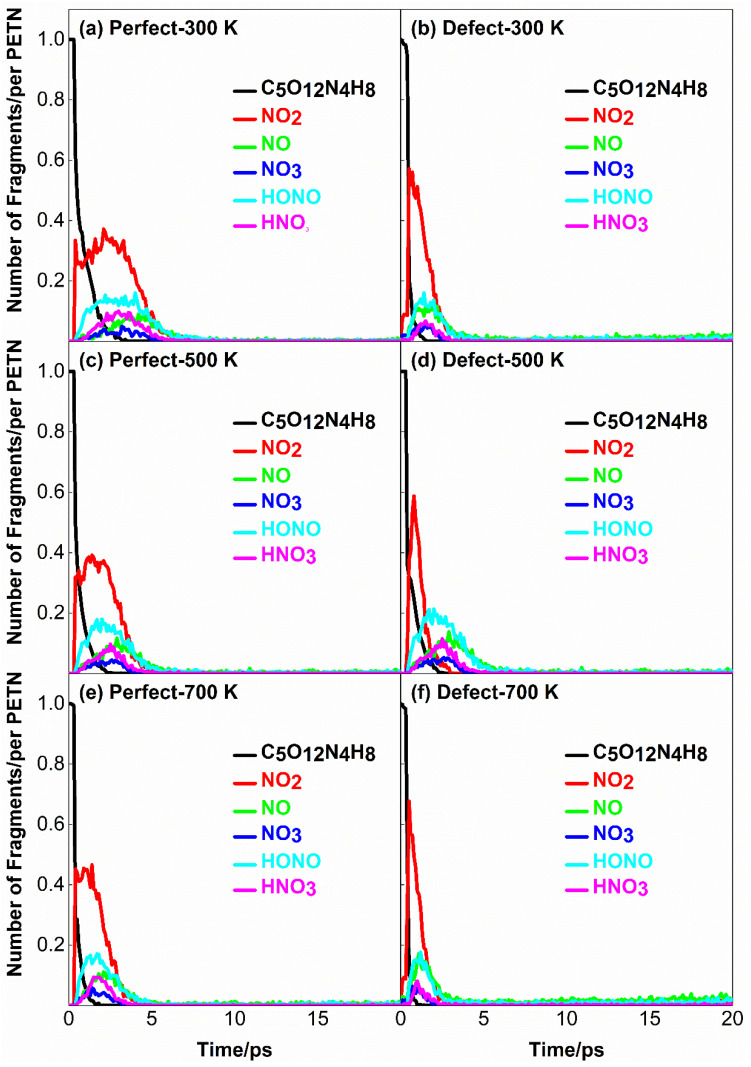
Time-dependent curves of initial decomposition products for the case of *U_s_* = 10 km·s^−1^. (**a**,**c**,**e**) correspond to perfect PETN crystal with preheating temperatures 300 K, 500 K, and 700 K; (**b**,**d**,**f**) correspond to defective PETN crystal with preheating temperatures 300 K, 500 K, and 700 K.

**Figure 8 molecules-28-02911-f008:**
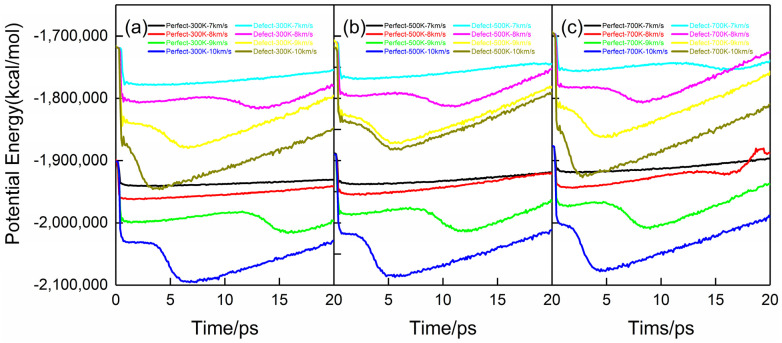
Time-dependent curves of potential energy for the case of *U_s_* = 10 km·s^−1^. (**a**–**c**) correspond to PETN crystal with a preheating temperature of 300 K, 500 K, and 700 K.

**Figure 9 molecules-28-02911-f009:**
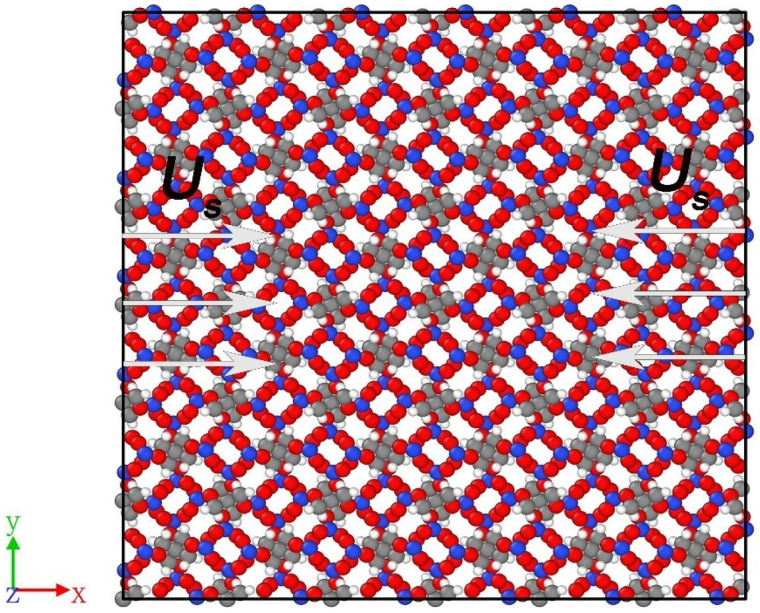
PETN supercell and shock compression model. C, H, O, and N are represented by gray, white, red, and blue spheres, respectively. As is the shock velocity, the arrows point to the shock onset directions, *U_s_* = 7 km·s^−1^, 8 km·s^−1^, 9 km·s^−1^, 10 km·s^−1^.

**Table 1 molecules-28-02911-t001:** Decomposition rate *k*(10^12^ s^−1^) of PETN molecules under the coupling effect of preheating, shock, and defect.

*U_s_* (km·s^−1^)	Preheating Temperature (K)	* **K** * _ **p** _	* **K** * _ **d** _	$\left[\left(k_{d}\;-\;k_{p}\right)/k_{p}\right]$ × 100%
7	300	0.00688	0.09167	92.5%
	500	0.05543	0.15448	64.1%
	700	0.07523	0.28369	73.5%
8	300	0.03000	0.33122	90.9%
	500	0.13643	0.46627	70.7%
	700	0.24234	0.74792	67.6%
9	300	0.26535	0.93081	71.5%
	500	0.45057	1.26102	64.3%
	700	0.77840	1.55408	50.1%
10	300	1.26635	1.90714	33.6%
	500	1.71408	2.25261	24.1%
	700	2.04698	2.29648	10.9%

## Data Availability

Not applicable.

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
