# Peer review of "Initial Response of Pentaerythritol Tetranitrate (PETN) under the Coupling Effect of Preheating, Shock and Defect via the Molecular Dynamics Simulations with the Multiscale Shock Technique Method"

_molecules, 2023, doi:10.3390/molecules28072911_

Round 1

Reviewer 1 Report (Previous Reviewer 5)

I have no particular objections to publication of this paper in its present form.

Author Response

Thank you for your affirmation of the paper. I will carefully check the grammar and spelling of the paper. Thank you!

Reviewer 2 Report (New Reviewer)

In the manuscript, authors make use of MD simulations to study the initial response of PETN under the effect of preheating, impact, and defects. Overall, the paper is well written but before publishing it requires some small changes and clarifications.

1.       Provide some more details about the MD simulation in the manuscript. For example, the minimization technique used and what type of potentials are used.

2.       I would recommend authors go through the labeling of the figures. For example, in line 202 it should be Figure 4, not Figure 7. Also, there are some spelling mistakes in the figures themselves.

3.       What is CJ?

4.       Quality of the figures can also be improved. 

Author Response

This manuscript is a resubmission of an earlier submission. The following is a list of the peer review reports and author responses from that submission.

Round 1

Reviewer 1 Report

In this paper, the initial response of PETN under the coupling effect was simulated by MSST method and molecular dynamics. The research is appropriately designed and the methods are adequately described. The research is comprehensive and the result is valuable. This submission is worthy of publication.

Author Response

Thank you for your affirmation of the paper. I will carefully check the grammar and spelling of the paper. Thank you!

Reviewer 2 Report

The authors present results from simulations in PETN using the multiscale shock technique to induce shock initiated chemical reactions. Effects from preheating and inducing a crystal defect are assessed.

There is a major flaw in the simulations that needs addressed, therefore I recommend rejection of this article.

In Figures 2 and 9, the authors show that both temperature and internal energy are rising at late times, and attribute this to "slow reactions and energy release", but if that were the case, internal energy would be continuing to decrease and higher energy intermediates form products. From other figures, it is clear that chemistry is effectively finished at these points. In this case, total energy is increasing, where it should be constant, as MSST has finished putting energy into the system and it should be roughly adiabatic.

ReaxFF has a known deficiency at high temperatures and pressures where total energy is not conserved. This causes increased temperature rises and artificially accelerates chemical reactions. This is an effect that would be more present in the stronger shock systems, causing the conclusions regarding increased kinetic rates, while still probably true, most likely highly overstated.

Reference #1 in the authors manuscript is one of numerous ReaxFF papers that lowers the timestep in the simulation to prevent this. For the strong shock cases here (9 and 10 km/s) a 0.025 or 0.01 fs timestep is needed to have defensible conclusions. Hence, this paper should be rejected to give the authors an opportunity to rerun the necessary simulations and re-assess results.

Other minor critics for the article include:

- the size of the void/defect should include number of molecules, as this size is in between the vacancy cluster and a true void.

-Figures 2 and 9 would be better if each plot had one Us and color in each was for preheat/defect such as in Figure 6

-Figure 3 and 7 are redundant information

-Figure 5 and 6 are redundant information, Figure 5 can be supplemental

-The calculated pressure values from the Us-Up results in section 3.4 do not match well with pressures in Figure 3, where the plotted pressure for 10 km/s is over 80 GPa, but from Us-Up you get 75 GPa

-The info in Figure 3 and 7 should come before all other results. The state of the shock (P, Us) should be characterized first, then the material response of the shock (T rise and reactions).

Reviewer 3 Report

The artile may be accepted in present form

Author Response

(The authors gave the same response as above.)

Reviewer 4 Report

In their manuscript, the authors present the results of the simulation of the initial response of PETN under the coupling effect of preheating, shock, and defect. The Multiscale Shock Technique combined with molecular dynamics was used for this study. It is obtained that the initial preheating and defects accelerate the decomposition of PETN, and the higher the preheating temperature, the faster the decomposition of PETN which, along with the list of the products. That could be addressed as the significant results of the study.

In the current form, I would not recommend the publication of the manuscript, because in my opinion there are some major open questions that should be addressed:

1.     There is no explanation for the decision that “PETN decomposed with the increase of system temperature, and the energy released further increased the temperature; The occurrence of secondary chemical reaction makes the temperature rise slowly”. The explanation is significant because the curves given in Fig.2 could be interpreted in different ways considering other results presented in the paper.

2.     Similarly, the evidence needed for the decision concerning “the pressure rise caused by a chemical reaction within 1~5 ps; After 5ps, the pressure decreases slowly, which is due to the expansion of the medium in the reaction process.”

3.     Some expressions must be rewritten because they are inaccurate. For example, “The amount of all initial products increases little with the increase of initial preheating temperature…”. The amount of the initial product itself could not be increased.  One may discuss the amount of product formed at certain time periods. This amount could be different due to decomposition reaction speed dependence on various physical and chemical parameters. It also necessarily considers that the reactant number in the crystal with and without defect is different.

And minor:

1.     The abstract of the paper is non-informative. It is written in a non-appropriate- form ( see Writing an Abstract for Your Research Paper – The Writing Center – UW–Madison (wisc.edu).

2.     Please check a sentence on 42-43 lines. It seems part of the sentence is missing.

3.     The MSST and MD abbreviations must be given where they are mentioned the first time.

4.     There is needed basement why the authors state that “the defect volume accounts for about 9% of the total PETN volume is close to the limit of 30% of the observed defect concentration”.

5.     The explanation of abbreviations NVT and NPT must be given.

6.     In the caption of Fig 3, c must be instead of x and the explanation of the meaning of the color must be added.

7.     The sentence “ … while the slowest decomposition of PETN molecules in defect-free PETN crystal at 300K preheating temperature.” is incomplete. (line 216).

8.     The non-defect crystal is called “without defect”, “defect-free” and “perfect”. Please choose one.

9.      Please pay attention to the water formulae (H2O) on line 268.

10.   The word is missed in this sentence “Compared with Figure 8(a) and Figure 8(b), the peak value of NO2 quantity in the defect containing system is 1.5 times that in the defect-free system…” (line 318).

11.  Please correct Eq. 3 ( line 379).

And to the end, in my opinion, this manuscript is much more related to the topic of the MDI journal Energies than that Molecules. The value of the study will be increased if the target group of scientists reads it.

Reviewer 5 Report

This is an interesting contribution to understanding the property of reactive materials under sudden stresses.

I have qualms however about the introductory presentation of the methodological approach, MSST + MD: no details are given, only bibliographic references. I think it would have appropriate to include a concise but more consistent presentation.

More serious, however, is the  lack of a clear and systematic  description of the observed chemistry in the system, and how it compares with experimental data.

For these reasons I think that the manuscript should undergo some revision, before being published.

A minor note: equations are badly formatted (eq. 3 could be avoided at all).

Round 2

Reviewer 2 Report

The authors have failed to address my major concerns in there misuse and misinterpretation of ReaxFF results. While any late stage reactions, like those seen here, could cause a temperature rise, the formation of products would lower the potential energy of the system. Yet, Figure 9 shows that potential energy is increasing. The only thing that could be causing this is poor energy conservation due to ReaxFF energy drift, which is a well know effect at high temperatures and pressures.

The authors note that reactions are still occurring and energy changing because the simulations only last 20ps, but if they ran them out longer, the energy would continue to rise until the simulation crashes.

All claims the authors make about late reactions causing energy rise are incorrect, and are caused by a known deficiency in the methods. A much smaller time step could help to prevent this, but not negate it.

If this manuscript is to be publishable, considerable changes need to be made that are outside the scope of a simple reply to the reviewers, including re-running and additional analysis on a majority of the simulations to better understand the main causes of the trends shown here. Hence, this paper should be rejected and re-submitted at a later date when the study has been properly conducted.

Reviewer 4 Report

The author does not pay attention to the following inaccuracies in their paper:

1. The abstract is not rewritten in the proper way.

2. There is no explanation for the decision that “PETN decomposed with the increase of system temperature, and the energy released further increased the temperature; The occurrence of secondary chemical reaction makes the temperature rise slowly.  

3. MSST abbreviation is given in the abstract without any explanation.

Eq. 3 is written incorrectly;  (-kt) is a degree.

And additionally, how is it possible, that the system is isothermal and isobaric? (see line 125). The definition “isobaric” and “isothermal” are used to identify processes.

How it is possible “The initial density of PETN with defects after optimization is 1.789 g·cm−3, while that of PETN with defects after optimization is 1.626 g·cm−3. “ (line 251)

Please change the formatting on line 298.

Please check the chemical formulas in the chapter of Fig. 2

Reviewer 5 Report

The authors made an attempt to recognize some of the referees' suggestions.  But after reading the new version of the manuscript - and the cover  letter stating that "As the time and space scales involved in this paper are small, it is difficult to observe the relevant phenomena with the existing means of implementation, and therefore it is difficult to compare with experiments" - I have very serious doubts that the exposed methodology aptly describes the phenomena addressed, especially for the known weaknesses, stressed by one of the referees, of ReaxFF.

But this is something the readers themselves can judge.